# How to Educate the Public about Dental Trauma—A Scoping Review

**DOI:** 10.3390/ijerph19042479

**Published:** 2022-02-21

**Authors:** Magdalena Nowosielska, Joanna Bagińska, Agnieszka Kobus, Anna Kierklo

**Affiliations:** 1Department of Gerostomatology, Medical University of Bialystok, 15-267 Bialystok, Poland; magdalena.nowosielska@umb.edu.pl; 2Department of Dentistry Propaedeutics, Medical University of Bialystok, 15-295 Bialystok, Poland; agnieszka.kobus@umb.edu.pl (A.K.); anna.kierklo@umb.edu.pl (A.K.)

**Keywords:** scoping review, traumatic dental injuries, dental trauma, education, knowledge

## Abstract

Educating the general population about dental trauma is of public health interest. The aim of this scoping review was to map research on traumatic dental injuries (TDIs) education in the general population and to identify the most relevant methods of knowledge transfer. PubMed, Scopus, Web of Science All Databases, reference lists, and grey literature were searched. Articles in English published between 2000 and 2020 were included. A total of 32 articles fulfilled inclusion criteria. The most frequently tested modality was lecture/seminar/workshop. Studies focused mainly on teachers and medical staff as target groups. Post-intervention evaluation showed an increase in knowledge. In long-term follow-up, a decrease in knowledge was found. The effectiveness of different modalities varied. Studies comparing single-modal and multimodal approaches did not confirm the effect of combined methods. Printed materials are a practical mode for laypeople. Lectures should be reserved for professions with high probability of coming into contact with a TDI victim. The Internet can be a promising tool to educate people. Educators have to choose the method of communication most appropriate for the target population. The education should include topics related to dental trauma prevention. Further research is needed to investigate the effectiveness of multimodal TDI education.

## 1. Introduction

Traumatic dental injuries (TDIs) of hard and soft tissues within and around the oral cavity are relatively common and occur as a result of accidents during sports or leisure activities, falls, blows by objects, interpersonal violence, and motor vehicle accidents. They represent a serious public health problem with an epidemiological dimension, cause numerous clinical and financial implications, and significantly impair the quality of each stage of life [1,2,3]. TDIs often coexist with other multi-organ injuries, and therefore their consequences are underestimated [4,5]. There are several habits that may predispose to dental trauma: chewing ice cubes or stationery items (paper clips, pen), opening bottles, cans, or package with teeth, or practising sports without adequate protection (face mask, mouthguard) [6]. Appropriate actions taken immediately after a trauma have a great impact on prognosis and treatment options. First aid measures are particularly important in the case of tooth avulsion because an immediate replantation or a proper transport of such tooth to a dental surgery significantly improves the prognosis.

The prevention and posttraumatic management methods should be widely known, but numerous studies have shown a low level of such knowledge in the general population as well as among teachers, sports coaches, and even among healthcare professionals, including dentists [4,6,7,8,9,10,11]. Therefore, effective and efficient information about dental trauma seems to be essential. Currently, more and more tools are used to promote healthy lifestyles and minimise destructive behaviour (social media, ambient marketing). At the same time, many tools used in the past have ceased to work or bring lower effects than expected [12,13,14]. An efficient educational campaign has to be widely accessible and cost-effective. Authorities planning campaigns on dental trauma should be aware of the advantages and limitations of different modes of information delivery to choose the most appropriate one for the target group [14]. For this reason, it seemed important to us to collect and analyse available knowledge of the effectiveness of methods and tools used to promote information about dental injuries in general population.

A scoping review was conducted to systematically map the research on dental trauma education in general population and to identify the most relevant methods of delivery of such knowledge. The results of this scoping review may help to develop best practices for educating the public about TDIs.

## 2. Materials and Methods

### 2.1. Methodological Framework

The scoping review of the literature was conducted according to the design of Arksey and O’Malley [15]. This method was chosen because the purpose of the present study meets the requirements of a scoping review: to summarise and disseminate research findings. In particular, this type of literature review aims to explore the nature of a research activity in a research area and to identify knowledge gaps. The scoping review is also appropriate to examine a diverse range of publications in a complex area and to allow syntheses of results from different studies [15,16,17].

The following steps were used: framing questions for a review, literature search, assessing the quality of studies, summarising the evidence, and interpreting the findings [15]. The Preferred Reporting Items for Systematic Reviews and Meta-Analyses extension for scoping reviews (PRISMA ScR) checklist was addressed to draft the protocol [18]. All members of the research team were involved in the protocol development.

### 2.2. Framing Research Questions

The following research questions were formulated: “What are the intervention options for dental trauma education in general population?” and “What are the outcomes and the effectiveness of different methods?”.

### 2.3. Inclusion and Exclusion Criteria

The inclusion criteria comprised peer-reviewed papers published between 2000 and 2020 where at least one educational method on dental trauma in general population was evaluated. Literature was restricted to interventional studies. The following papers were excluded: books, book chapters, symposium proceedings, systematic or scoping reviews, meta-analyses, meta-syntheses, qualitative studies, editorials, letters to editor, and press releases. Another exclusion criterion was papers evaluating dental professionals (dentist, students of dentistry, dental hygienists, and assistants) being the only study group. Papers in which dental staff were one of the evaluated groups were included, but data on dental professionals were excluded from the analysis. Articles published in any language other than English or published before 2000 were also excluded. Papers that did not fit into the conceptual framework of the study were also excluded.

### 2.4. Information Sources and Search Strategy

The following bibliographic databases were searched: PubMed, Scopus, and Web of Science All Databases. The preliminary search was done in March 2020, the final search-on 31 March 2020. The search strategy was drafted by two authors (M.N. and J.B.) and then discussed and approved by all research team members. The final search strategy for PubMed included the following key terms: (((“tooth avulsion”[MeSH Terms] OR “tooth injuries”[MeSH Terms]) OR “tooth fractures”[MeSH Terms]) OR “traumatic dental injury”[All Fields]) OR “dental trauma”[All Fields]) AND ((“knowledge”[MeSH Terms] OR “education”[MeSH Terms]) OR “learning”[MeSH Terms]) AND ((“2000/01/01”[PDAT]: “2020/12/31”[PDAT]) AND English[lang]). In addition, a manual search of the reference lists of retrieved articles and grey literature (using the Google search engine) was conducted to identify additional relevant studies.

### 2.5. Study Selection

The results were exported into EndNote Web reference manager and duplicates were removed, with further duplicate removal when found later in the process. Titles and abstracts were screened independently by two reviewers (M.N. and J.B.) to identify the relevant articles. The other two authors (A.K.1 and A.K.2) were consulted when there were differences of opinion on whether an article met the inclusion/exclusion criteria. The number of such records was limited.

### 2.6. Data Extraction

A data extraction form was created by M.N. and reviewed by all research team members. It included the following information: geographic location, education method, study design, population, kind of intervention, term of observation, and main findings. The data extraction tool ensured a standardisation of data extraction and charting across the team [17].

Data extractions were performed independently by two reviewers (J.B. and A.K.1) and recorded with the data extraction form. Initially, five records were compared to the standardised data extraction through discussion with the whole research team. Subsequently, both researchers provided charting of all records, and the results were compared by the third researcher (M.N.). If any differences were found, they were discussed and a consensus was reached.

## 3. Results

The database search resulted in 3771 records (PubMed 303, Scopus 2.216, Web of Science All Databases 852). A manual search revealed an additional paper. After rejecting duplicates, 2533 records remained. The analysis of titles and abstracts resulted in 2499 articles classified as not relevant to this scoping review. Thirty-four papers met the study criteria, but at this stage, two papers had to be excluded due to the unavailability of full text. Finally, 32 studies were included for data extraction (Figure 1). Table 1 presents a summary of the studies that addressed TDI education and Appendix A presents full summary of studies focused on education in TDI [19,20,21,22,23,24,25,26,27,28,29,30,31,32,33,34,35,36,37,38,39,40,41,42,43,44,45,46,47,48,49,50].

### 3.1. Geographical Distribution

The geographical distribution of the records revealed that 40.6% (*n* = 13) were from the Middle East region (Israel [21,26], Kuwait [22,23,44], Iran [33,37,38,50], Turkey [28], Lebanon [32,45], United Arab Emirates [46]), 18.7% (*n* = 6) from India [27,34,39,43,48,49], 12.5% (*n* = 4) from Brazil [25,40,42,47], 6.3% (*n* = 2) from Hong Kong [35,36], USA [21,41], and Poland [30,31], and 3.1% (*n* = 1) from Tanzania [19], Switzerland [24], and Australia [29]. Most papers included in this scoping review (*n* = 23, 71.9%) were published between 2012 and 2018 [27,28,29,30,31,32,33,34,35,36,37,38,39,40,41,42,43,44,45,46,47,48,49].

### 3.2. Subjects

Most of the records targeted homogenous groups, but there were also several studies where different professions were evaluated (teachers/teaching assistants and school nurses; teachers, bank employees, and medical staff; teachers, pupils, and parents; school nurses and physical education teachers). Teachers were the most frequently evaluated group, accounting for 62.5% (*n* = 20) of the studies [19,20,21,24,25,27,28,31,34,35,37,38,39,43,44,45,46,48,49,50]. This number included four (12.5%) papers [21,25,39,46] identifying physical education teachers or sports coaches as a target population and one (3.1%) identifying physical education students [31]. Non-dental medical staff included physicians, paediatricians, nurses, and paramedics, as well as medical, nursing, and speech therapy students. The total number of studies concerning medical professionals was seven (21.9%) [20,25,29,30,42,46,47] of which three articles evaluated the effect of educational intervention with school nurses [20,30,46]. Four studies (12.5%) focused on parents [23,33,39,41] and three (9.4%) focused on pupils at the age between 7 and 20 [32,36,39]. The other groups were bank employees [25], military recruits [26], and firefighters [40]. The quantity of the intervention groups differed from 30 to 655 subjects, with a mean of 206.1 ± 170.9.

### 3.3. Intervention

Two different approaches were adopted during the intervention: the use of a single educational method [21,22,23,24,25,26,27,28,29,30,32,34,35,36,38,40,43,47,49] or a multifaceted approach—a combination of two or three different methods [31,33,37,39,41,42,46]. In some studies, authors compared the effectiveness of different modes of education [19,20,32,41,48,50]. The most frequently tested modality was lecture/seminar/workshops followed by a short discussion. It was used in 13 (40.6%) studies as a single method [21,22,25,26,27,29,30,34,40,43,44,47,50] and in 7 (21.9%) studies as a combination with some printed materials (posters or brochures/leaflets) [19,20,37,39,42,45,46] or with an application on mobile devices [44]. Emerich et al. [31] assessed the effect of lecture memorisation through the subjects’ own work when preparing a presentation for primary school students based on published literature on dental trauma. Eight papers evaluated the effectiveness of a single printed material: posters (*n* = 4, 12.5%) [24,35,36,38] and brochures/leaflets (*n* = 4, 12.5%) [19,20,23,28]. One compared the effectiveness of a poster vs. an online application [41] and another vs. a lecture [50]. Other types of intervention included the audio-visual mode: the authors presented a cartoon movie for children [32], a DVD presentation [49], and one study did not specify the kind of audio-visual intervention [48].

The content of most educational interventions covered the causes of TDI, symptoms, first aid, and management. However, 13 records (40.6%) [19,20,21,22,23,25,32,33,39,40,42,43,44] only presented information related to tooth avulsion. The materials were mostly prepared by the authors, some studies [41,44,46] used a poster and an application available at the International Association of Dental Traumatology website.

### 3.4. Knowledge Gain Assessment

The most common way to assess the effectiveness of a single educational method was a pre-test post-test study design without a control group, it was used in 17 studies (53.1%) [21,22,25,27,28,29,32,34,37,39,40,42,43,46,47,48,49]. Another way was to compare the knowledge of a study group after an intervention with the knowledge of a control group (subjects with no intervention) found in 9 records (28.1%) [23,24,26,30,33,35,36,38,45]. Some studies (n = 7, 21.8%) focused on comparing the effectiveness of different modalities (more than one experimental group) by comparing their knowledge gain after the intervention (with or without controls) [19,20,31,32,41,44,50].

### 3.5. Term of Observation

The evaluated studies differed in the time interval between the intervention and the final follow-up. Some authors assessed the knowledge gain immediately after the intervention (*n* = 8, 25%) [22,26,32,41,44,47,48,49] or one week later (*n* = 4, 12.5%) [23,33,35,36]. In four studies (12.5%) [25,28,38,43], post-intervention assessment was conducted between one and two months. The most common follow-up period was between three and six months (*n* = 12, 37.5%) [19,20,27,29,32,34,39,40,42,45,46,50]. The longest periods were as follows: ten months [21], one year [31], two [30], three [37], and five years [24], each taken in one study (3.1%). In another study, authors did not explicitly define the period of observation [39].

### 3.6. Key Findings

Records included in this scoping review indicated a low and insufficient level of knowledge of evaluated populations before the intervention. Post-intervention assessments showed an increase of knowledge, however, some authors reported that a sufficient level was not reached [21,23,26,33,34,36]. The repetition of messages increased the number of correct answers suggesting the need for frequent updating of information [32]. Long-term follow-up showed a decrease in information retention with time [37,40,42].

The effectiveness of particular modalities varied, and the data from evaluated reports were not consistent. In the study by Kahabuka et al. [19], a seminar was more effective than mail guidelines, but in another study, leaflets were more effective than a lecture [50]. Al-Musawi et al. [44] found that the Dental Trauma App was more effective than a lecture and even more effective than combining the application with a lecture. On the other hand, Iskander et al. [41] found a similar effect of a poster and a mobile application.

Some authors found the combination of lecture and printed materials to be effective [37,39,42,45,46], but studies comparing single and multimodal approaches did not confirm such effect [20,31,41,44]. According to McIntyre et al. [20], the TDI lecture provided no additional benefit to the educational pamphlet; similar observations were made when the lecture and the Dental Trauma App were combined [44].

## 4. Discussion

Prevention methods and first aid measures in dental trauma should be common knowledge. The unpredictable nature of such injuries and a high link between actions taken at the scene and the outcome of treatment indicate the need for dental trauma education campaigns. The aim of this scoping review was to summarise and disseminate the results of studies on educating the public about dental trauma. The results may assist policy makers and practitioners in choosing the most appropriate way to address the target population.

Thirty-two papers were successfully identified that met the inclusion criteria. This indicates that such education is of research interest, particularly in Asia and the Middle East. In contrast, no papers were found from regions and countries with developed oral health programmes, such as Scandinavia, Canada, or the United Kingdom. Systematic reviews of knowledge on TDIs by Tewari and co-authors also revealed discrepancies in the geographical distribution of studies [9,10,11].

Our main observation is that a wide range of educational modalities was tested for their effects and that the studies differed in terms of the groups studied, the follow-up period and the results obtained. Another important finding is that substantial knowledge gains were observed in every study evaluated in this review, regardless of the subjects and the applied tool. However, none of them found that all participants were able to assimilate the full range of information presented to them. The most favourable outcome of dental trauma education is long-term maintenance of knowledge and improved attitudes and willingness to provide first aid to the victim. In several studies, the post-test was conducted immediately after intervention, therefore, the effectiveness of the methods evaluated was poorly validated. However, longitudinal observations indicated that knowledge retention was still observed after several months and even years, with decrease in some areas. According to Custers [51], between two-thirds and three-fourths of knowledge acquired in school would be retained after one year, falling to less than fifty percent in the following year. Data from the study with the longest observation period between the intervention and the follow-up were presented by Lieger et al. [24]. They found that a poster campaign in the population of school teachers was sufficient for adequately maintaining knowledge for five years. However, due to a very low response rate in the intervention group, these results should be interpreted with caution. It is highly likely that only those respondents who were confident in their knowledge responded to the survey.

There seems to be no universal educational method suitable for every population. When planning an informative campaign, cost-effectiveness in crucial. Before making a decision, the accessibility of the target group, the literacy level of population, the socio-economic status, e.g., the access to modern technologies or the Internet, the availability of staff trained in dental trauma, and the source and size of funds have to be assessed.

For the general population, the following methods should be considered: printed materials, web contents, and mass media. Printed informative brochures, leaflets, pamphlets, and posters are traditional tools for delivering knowledge to a wide audience, and their effectiveness in dental trauma education has been well documented. The cost of printed educational materials depends on their quality and quantity, but they are relatively inexpensive to produce and distribute [35,36]. A significant advantage is that such manuals are available for free from several sources, for example on the IADT website [41]. Nevertheless, conducting a nationwide campaign requires considerable financial resources [51]. The main disadvantage of printed materials is the limited space to provide complete information [23]. Moreover, due to the layout of an educational tool, there may be difficulties in determining the best possible action [41]. Information conveyed through printed materials may not be fully assimilated by the target group due to selective reading or incomplete understanding of given knowledge [36]. In public places, such as schools or hospitals, posters seem to be more appropriate than other printed materials. People tend to lose leaflets, whereas posters can be constantly accessible and conspicuous [39]. The only limitation is that some institutions frequently change the materials displayed on notice boards or information walls [35,36].

Currently, the Internet is laypeople’s preferred source of knowledge including on medical topics [52,53]. Mobile healthcare applications can be a way to communicate efficiently in dentistry [14,41,54]. The widespread accessibility of mobile devices (cell phones, smartphones, tablets) increases the availability of appropriate first aid procedures in dental trauma even at the scene of an accident [41]. Surprisingly, the limited number of papers evaluating the impact of using modern technologies such as educational tools in dental trauma was found [41]. Moreover, the results are inconclusive, especially those obtained by Al-Musavi et al. [44]. Participants using the Dental Trauma App as an educational tool scored higher than those attending a lecture, but those who were educated with both methods scored the lowest. In contrast, Iskander et al. [41] found that a poster and a mobile application were similarly effective in delivering dental trauma information. They also found that lower educated parents were more satisfied with the information from the mobile application than from printed materials. According to Jabocs et al. [55], the Internet should not be regarded as a substitute for other sources of health information, as this may exclude certain groups from accessing the health information necessary for making health decisions. The effectiveness of online applications depends on many aspects, such as being familiar with the use of smartphones, the type of device (screen without enlarging screen function), personal preferences, or the difficulty of using the application in a stressful emergency situation [44]. In developing countries, people may have problems with the online health education due to slow connection speeds or other technical problems or even due to the lack of Internet access [54]. For such societies, more traditional educational methods such as leaflets may be more valuable [54].

Although social media can be a promising tool for dental trauma education, this content has to be critically appraised in terms of the reliability of presented materials. Evidence-based information on dental trauma was applied in the studies evaluated, but there are reports showing that the online resources are of uncertain quality. Hutchinson et al. [56], after a detailed analysis of YouTube^TM^, concluded that the materials available there on tooth avulsion were of a low quality. They even suggested that dentists should warn patients of the misleading information available on YouTube™ videos. Abu-Gazaleh et al. [57] found, based on an evaluation of Facebook content, that there was a limited number of posts about educational materials for laypeople. They mainly focused on using mouthguards. Topics on first aid and long-term management or prognosis in TDIs were missing. The acute nature of such accidents may be the reason why such content results in a lower involvement than other medical posts [57]. Abu-Gazaleh et al. [57] suggested to create interactive posts with pictures and clinical cases to attract a multitude of users.

The other mass media (tv, radio, newspapers, and magazines) still offer the widest possible exposure and are very effective in raising public awareness. The media can be an efficient tool for the health promotion, provided that pertinent requirements are met [51] A multimedia approach, including video, graphic design, and audio voice-over, can also be used to convey information to people with limited literacy skills [9]. This mode of public communication has the advantage of spreading knowledge quickly and reaching the largest possible audience [29]. However, paid advertisements, especially on television, can be very expensive, therefore the costs and benefits need to be carefully weighed with regard to the education on dental trauma [51]. No studies were found on the role of traditional mass media (television, radio, printed newspapers, and magazines) as a source of knowledge of dental trauma in adults. The only study using a cartoon movie was conducted in a group of 8–11-year-old Lebanese children [31]. The authors found that for this population, the audio-visual method was a good way to convey knowledge on tooth avulsion. Comparing auditory and visual modes used separately, the first showed a greater impact on knowledge retention. When the two modes were combined, watching a movie twice was more effective than listening to a story twice. Moreover, children preferred movies to oral stories. This study also highlighted the importance of repetition in knowledge transmission, which is very easy via mass media, for example in the form of a short movie played several times before a popular children’s programme. The cost of such broadcast should be covered by national and local government agencies or even commercial companies [31].

There are some specific groups that should be educated in dental trauma prevention and first aid, including school staff and sports coaches. Children and adolescents are particularly vulnerable to TDIs, with approximately 19% of TDIs occurring at schools [58]. The scoping review of literature presented here showed that most studies addressed this audience, with teachers most frequently being the target group. Medical professionals, especially those who deal with emergency management or work with children, such as school nurses, are another group that is highly likely to encounter patients with dental trauma [6]. It was reported that there was a gap in the non-dental medical staff curriculum and that medical doctors showed a low level of competence in the management of dental emergencies [7]. For both abovementioned groups, education through personal contact with the educator (lecture, seminar, workshops) may be the most effective way to enhance knowledge and competences. The opportunity for the audience to ask questions and the doubts expressed in discussion increased the efficacy of information transfer [14]. Multimodal workshops were highly effective in improving self-assessed competences of emergency service providers and changed their routine in the treatment of dental emergency patients [29]. The “contamination effect”, e.g., the ability of teachers who attended a seminar on dental trauma management to instruct their colleagues on the proper behaviour, was observed [19,21]. A limitation of lectures is that they are expensive, time-consuming, and require many instructors to target an entire community (e.g., all teachers). Al-Asfour et al. [21] suggested that categories of staff other than dentists should be trained to present such information. When planning such educational campaigns, it is worth considering professional conferences, which allow a large number of participants to be reached [21,29]. An alternative to a lecture could be an interactive video because it does not require the presence of a lecturer, can be played many times, and can be stored for a long time [49]. Such a solution seems particularly beneficial in times of limited access to training, such as during the ongoing COVID-19 pandemic. However, the simplest way to educate medical staff is to include TDI topics in their curriculum [37].

Our scoping review of literature has shown that there is no evidence that the multifaceted approach is more effective than a single method [20,31,44]. The purpose of complex interventions is to work in synergy [51], and it might be expected that the memorisation effect would be present in those subjects who received the knowledge in more than one way. In fact, two studies found that combining a lecture with another activity (preparing presentation or using smartphone applications) was less effective than a lecture alone [31,44]. If this finding is true, many costs could be saved in education campaigns by focusing on only one modality. However, “the absence of evidence is not evidence of absence”, therefore, this result should be carefully interpreted. This aspect of education on dental trauma definitely requires further investigation, especially in the light of developing technology and educational platforms that could be more suitable for young children and adolescents, in addition to standard lectures.

One of the limitations found when interpreting the findings is that the results of studies without a control group may be skewed. Respondents may pay particular attention to certain familiar themes after answering the questionnaire at baseline [26]. This may be the case especially in short-term assessment where the time intervals between the pre-test, the intervention and the final assessment are short, which may result in a higher proportion of correct answers. The problem with the short follow-up period adopted in the methodological framework in many studies has been explained above.

Our scoping review revealed some directions for future research. As previously mentioned, the comparison of single and multifaceted approaches seems to be a very important aspect to investigate. Research designs should include longitudinal observations because the short-term effect of different modalities has already been widely evaluated. Topics related to the prevention of dental trauma should be included, and the assessment of knowledge gained in this area should be included in the evaluation. The educational effect of the Internet and social media contents has to be analysed. Papers included in this scoping review mostly used questionnaires in their pre-test-post-test study design. As clinical practitioners, we should consider the extent to which such study design reflects the real benefit of dental trauma education, i.e., how effective such education is in improving post-TDI outcomes. Hence, it would be interesting to relate the level of the awareness of TDI prevention, as reflected in survey results, with the incidence of TDI and their complications in a given population.

A scoping review as a method of literature review has some limitations as no quality analysis of included papers is performed [16]. Some studies included in this analysis presented poor data quality (lack of properly explained methodology), which could have increased the risk of bias. They were included because scoping reviews aim at providing a comprehensive overview of all the literature related to the search area without synthesizing evidence from different studies [15,16,59]. However, a systematic review and meta-analysis of available papers would provide clearer data. Another limitation of the present scoping review is the inclusion of papers written in English only, however, it is a common practice in this type of review. There are also new papers published after the literature search had been done [60,61], which is why they were not included in the analysis.

## 5. Conclusions

Educating the general population about dental trauma is in the interest of public health. Due to its low costs and wide access, educational printed materials seem to be the most practical way for laypeople. The education mode involving the personal presence of an educator (lectures and workshops) should be reserved for those target groups for whom the probability of TDI management is very high and who are easily accessible (teachers, sport coaches, non-dental medical staff). Educational campaigns in mass media are expensive and external resources need to be involved. Online sources can be a promising tool to educate people, especially for those familiar with modern technologies, but awareness campaigns are still needed to get potential users to upload them and to prepare for appropriate behaviour in the event of an accident. In TDI education, costs and potential outcomes have to be carefully considered, and educators have to choose communication methods that are most suitable for the target population. The education should cover dental trauma prevention topics, and the knowledge gained should be included in the evaluation. Further research is needed to investigate the effectiveness of combining different methods of knowledge transfer, e.g., the combined effect of a lecture and a leaflet or an online application and a poster.

## Figures and Tables

**Figure 1 ijerph-19-02479-f001:**
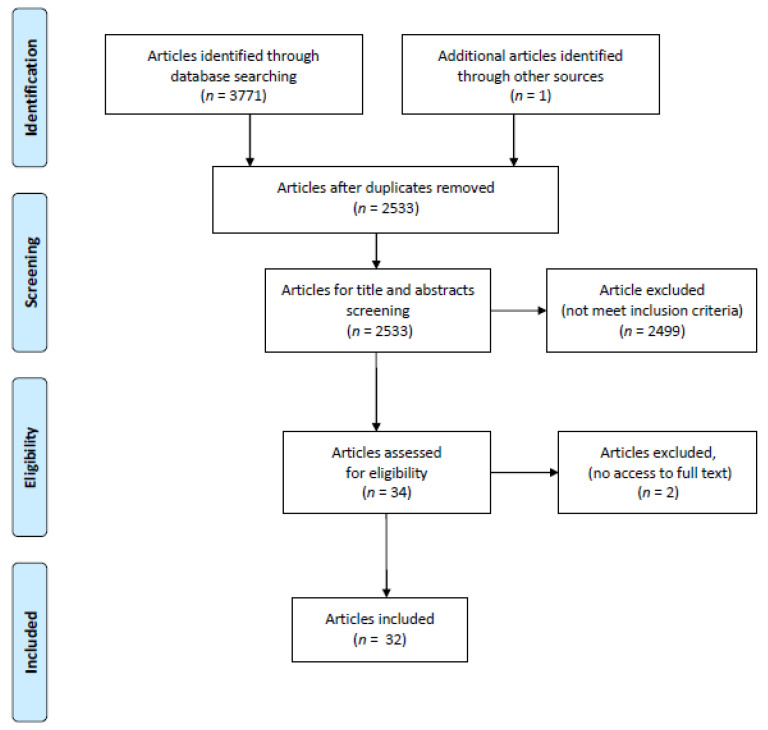
Literature search flow diagram.

**Table 1 ijerph-19-02479-t001:** Summary of studies focused on education in TDI.

Authors	Year	Country	Modality	Sample	Term of Observation
Kahabuka et al. [19]	2003	Tanzania	a seminar vs. a guidelines sent by email	teachers	6 months
McIntyre et al. [20]	2006	USA	a pamphlet vs. a pamphlet and a 10-min lecture	public elementary school teachers, teacher’s assistants, nurses	3 months
Holan et al. [21]	2006	Israel	a seminar	physical education teachers	10 months
Al-Asfor et al. [22]	2008	Kuwait	a lecture	teachers	after intervention
Al-Asfor, Andersson [23]	2008	Kuwait	a leaflet	parents	1 week
Lieger et al. [24]	2009	Switzerland	a poster	teachers	5 years
Frujeri, Costa [25]	2009	Brazil	a lecture	different professionals	2 months
Levin et al. [26]	2010	Israel	a lecture	18-year-old men military recruits	after intervention
Karande et al. [27]	2012	India	a lecture	teachers	3 months
Arikan, Sȍnmez [28]	2012	Turkey	a leaflet	teachers	1 month
Skapetis et al. [29]	2012	Australia	an interactive and multimodal workshop	physicians, nurse practitioners, medical students	6 months
Baginska, Wilczynska -Borawska [30]	2012	Poalnd	a lecture	school nurses	2 years
Emerich et al. [31]	2013	Poland	a lecture + an additional task	physical education students	1 year
Soubra, Debs [32]	2013	Lebanon	audio and visual methods	pupils (8–11 years old)	3 months, after intervention
Ghaderi et al. [33]	2013	Iran	a leaflet	parents	1 week
Pujita et al. [34]	2013	India	a lecture	teachers	3 months
Young et al. [35]	2013	Hong Kong	a poster	teachers	up to 1 week
Young et al. [36]	2013	Hong Kong	a poster	pupils (11–20 years)	up to 1 week
Raoof et al. [37]	2014	Iran	a multimodal educational intervention (lecture + poster)	health teachers	3 years
Ghadimi et al. [38]	2014	Iran	a poster	school health teachers	1 month
Grewal et al. [39]	2015	India	a multimodal educational intervention (flip cards + a poster + interactive sessions and a lecture)	school teachers/sport coaches, pupils (7–12 years old), parents	6 months
Perazzo et al. [40]	2015	Brazil	a lecture	firefighters	6 months
Iskander et al. [41]	2016	USA	a mobile application vs. a poster	parents	after intervention
Cruz-da-Silva et al. [42]	2016	Brazil	a multimodal educational intervention (a lecture + a pamphlet)	non-dental health professional involved in the emergency care	6 months
Taranath et al. [43]	2017	India	a Power Point presentation	school teachers	1 month
Al-Musawi et al. [44]	2017	Kuwait	a lecture vs. a lecture + access to the Dental Trauma App vs. access to the Dental Trauma App	female teacher	after intervention
Yordi et al. [45]	2017	Labanon	a multimodal educational intervention (a PowerPoint presentation + a brochure + a poster)	teachers	6 months
Al Sari et al. [46]	2018	United Arab Emirates	a multimodal educational intervention (a workshop + a poster)	school nurses and physical education teachers	3 months
Nagata et al. [47]	2018	Brazil	a lecture	health courses’ students	after intervention
Nashine et al. [48]	2018	India	audio vs. audio-visual aids (not specified)	teachers	after intervention
Niviethitha et al. [49]	2018	India	an interactive educational DVD video	teachers	after intervention
Razeghi et al. [50]	2019	Iran	a leaflet vs. a lecture	teachers	6 months

## Data Availability

Data available on request from the authors.

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
