# Peer review of "How to Educate the Public about Dental Trauma—A Scoping Review"

_ijerph, 2022, doi:10.3390/ijerph19042479_

Round 1
Reviewer 1 Report
The purpose of the scoping review was to systematically map the research on dental trauma education in general population and to identify the most relevant methods of delivery of such knowledge. However, there are some flaws which reduce the reliability of conclusions-
- It is a scoping review with limitations in terms of time and language. This is not the ideal method of evidence based dentistry. Rather, a systematic review with thorough analysis of literature must have been performed.
- The Literature was last searched about 2 years ago. This makes the contents of the literature obsolete. The literature search strategy has not been highlighted as well.
- Since it is a scoping review, the quality analysis has not been performed. Lack of assessment of risk of bias and its effect upon the interpretation of the results is a major short coming which reduces the significance.
Author Response
We would like to thank the Reviewer for the valuable comments and recommendations.
The purpose of the scoping review was to systematically map the research on dental trauma education in general population and to identify the most relevant methods of delivery of such knowledge. However, there are some flaws which reduce the reliability of conclusions
- It is a scoping review with limitations in terms of time and language. This is not the ideal method of evidence based dentistry. Rather, a systematic review with thorough analysis of literature must have been perform. Response: Our aim was to identify as many as possible concepts used for dental trauma education and to discuss them. But we agree that a systematic review should be conducted to confirm obtained results. It has been now underlined in the paragraph referring to the study limitations.
- The Literature was last searched about 2 years ago. This makes the contents of the literature obsolete. The literature search strategy has not been highlighted as well.
Response: Indeed, the time between performing the literature analysis and preparing the paper for publication was relatively long. This was mainly due to new responsibilities at the university and limitations in direct contacts due to the COVID 19 pandemic. In our opinion, this scoping review comprehensively provides an overview of the work on this topic. Information that new publications (not included in the analysis) were published after the literature search had been done, has been added to the limitations.
3. Since it is a scoping review, the quality analysis has not been performed. Lack of assessment of risk of bias and its effect upon the interpretation of the results is a major short coming which reduces the significance.
Response: We agree that the lack of quality analysis of papers included in our review may have increased the risk of bias. We underline this fact in the last paragraph of Discussion.
Reviewer 2 Report
Although your research topic, Education on Dental Trauma, is interesting, some revisions are required.
Kindly ensure your manuscript is more comprehensive by making the following revisions.
1. The study used papers published between 2000 and 2020. Could you explain why the scope of the study starts from 2000?
2. Line 135: If this line means that the studies that met the study criteria conducted education on dental trauma between 2012 and 2018, kindly provide more precise descriptions to clarify your meaning.
3. The frequency results are required for the 32 papers selected for this study.
4. Merely outlining and summarizing the study results is significantly limiting. Meta-analysis of the intervention studies would provide clearer results. Consider this an option to improve the quality of your study.
If there was a particular reason why meta-analysis was not conducted, please explain it in the study limitations section of your manuscript.
5. Line 60, 267, 269, 290: Parentheses are required for references.
6. Line 408: More detailed descriptions are required.
Author Response
We would like to thank the Reviewer for the valuable comments and recommendations.
Although your research topic, Education on Dental Trauma, is interesting, some revisions are required.
Kindly ensure your manuscript is more comprehensive by making the following revisions.
- The study used papers published between 100 and 2020, could you explain why the scope of the study starts from 2000? Response: The initial literature review did not identify papers on this topic published before 2000. We therefore decided to limit the number of papers for review to the period 2000-2020.
- Line 135: If this line means that the studies that met the study criteria conducted education on dental trauma between 2012 and 2018, kindly provide more precise descriptions to clarify your meaning. Response: This sentence expresses our conclusion that dental trauma education received the most attention from researchers between 2012 and 2018, as most papers included in the analysis were published during this period. In this way, we wanted to emphasise that the topic was continuously relevant. The sentence has been rewritten to better express our observation.
- The frequency results are required for the 32 papers selected for this study. Response: The frequency of results has been check and additional values have been presented (paragraph 3.4)
- Merely outlining and summarizing the study results is significantly limiting. Meta-analysis of the intervention studies would provide clearer results. Consider this an option to improve the quality of your study.
If there was a particular reason why meta-analysis was not conducted, please explain it in the study. Response: We agree that a systematic review and a meta-analysis would provide clearer results. It has been now underlined in the paragraph referring to the study limitations. However, our aim was to identify as many as possible concepts used for dental trauma education and to present to the readers how such research was conducted. - Line 60, 267, 269, 290: Parentheses are required for references. Response: It has been corrected.
- Line 408: More detailed descriptions are required. Response: This conclusion has been rewritten.
Reviewer 3 Report
Thank you for the opportunity to review this manuscript. The scoping review is relevant to dentistry as it assesses educational strategies for the prevention of dental trauma.
The manuscript presents a topic relevant to dentistry at a public health level that is the prevention of dental trauma in childhood.
The authors performed a very well-designed scoping review to verify studies that evaluate dental trauma bristle education strategies for non-dentists.
The approach is interesting since those who assist children in trauma situations, at first, are parents or other professionals.
The text is very well written and the conclusions are consistent with the findings. The review was very designed and described. The results are clear and well discussed.
I suggest only describing the acronym “TDI” in full the first time it appears in the abstract.
Author Response
We would like to thank the Reviewer for the favourable assessment of our paper.
Thank you for the opportunity to review this manuscript. The scoping review is relevant to dentistry as it assesses educational strategies for the prevention of dental trauma.
The manuscript presents a topic relevant to dentistry at a public health level that is the prevention of dental trauma in childhood.
The authors performed a very well-designed scoping review to verify studies that evaluate dental trauma bristle education strategies for non-dentists.
The approach is interesting since those who assist children in trauma situations, at first, are parents or other professionals.
The text is very well written and the conclusions are consistent with the findings. The review was very designed and described. The results are clear and well discussed.
I suggest only describing the acronym “TDI” in full the first time it appears in the abstract.
Response: It has been corrected.
Reviewer 4 Report
Dear authors,
Thank you for submitting your manuscript and for your effort in conducting this research.
I believe this manuscript is important and lead to valuable insights regarding education about TDI and a scoping review can show the gaps that need to be fulfilled.
I found only three minor concerns in your manuscript. The first is the length of the manuscript itself, considering the table. Since your review had two questions to be answered and several topics, I understand that the manuscript should be longer. But I believe a shorter version of Table 1 or dividing the information into smaller charts could benefit the reading process, and the more complete table could be sent as supporting material..
The other minor issue I found was a few mistypes, especially in the Discussion topic when showing the citation numbers.
The last commentaries are to change the word "helmet" (line 35) for "face mask" or some other similar word - to show the idea of protective equipment for the maxillofacial structures; and on line 258, to give proper citation to the statement "...and their effectiveness in dental trauma education has been well documented."
Overall I believe that by changing these minor issues, this manuscript should be accepted for publication.
Author Response
We would like to thank the Reviewer for the the favourable assessment of our work and valuable comments.
Dear authors,
Thank you for submitting your manuscript and for your effort in conducting this research.
I believe this manuscript is important and lead to valuable insights regarding education about TDI and a scoping review can show the gaps that need to be fulfilled.
I found only three minor concerns in your manuscript. The first is the length of the manuscript itself, considering the table. Since your review had two questions to be answered and several topics, I understand that the manuscript should be longer. But I believe a shorter version of Table 1 or dividing the information into smaller charts could benefit the reading process, and the more complete table could be sent as supporting material.
Response: Table 1 has been shortened and rearranged. The complete table has been submitted as a supplementary material.
The other minor issue I found was a few mistypes, especially in the Discussion topic when showing the citation numbers.
Response: It has been corrected.
The last commentaries are to change the word "helmet" (line 35) for "face mask" or some other similar word - to show the idea of protective equipment for the maxillofacial structures; and on line 258, to give proper citation to the statement "...and their effectiveness in dental trauma education has been well documented."
Response: It has been corrected.
Overall I believe that by changing these minor issues, this manuscript should be accepted for publication.
Round 2
Reviewer 1 Report
The flaws highlighted in the last review have not been addressed. The authors have tried to justify their review methods which has the deficiencies as per my opinion.
Author Response
The flaws highlighted in the last review have not been addressed. The authors have tried to justify their review methods which has the deficiencies as per my opinion.
Response
We regret to note that both, the changes made to the content of the article and our explanations in response to the Reviewer's comments, were not accepted. We would like to emphasise once again that the choice of a scoping review as a method was deliberate as in this way it was possible to cover the widest range of publications on the topic. We also believe that despite the fact that several months have passed since the literature review was conducted, the paper still has research value. The two additional papers that were published on this topic would only account for 6% of the total articles assessed.
As suggested by the Reviewer, the manuscript was proofread by a professional English translator with many years of experience in proofreading medical scientific articles.